# Structure of MHC class I-like MILL2 reveals heparan-sulfate binding and interdomain flexibility

Mizuho Kajikawa[1,2], Toyoyuki Ose[3], Yuko Fukunaga[2], Yuki Okabe[2,3], Naoki Matsumoto[4], Kento Yonezawa[5], Nobutaka Shimizu[5], Simon Kollnberger[6], Masanori Kasahara[7] & Katsumi Maenaka [2,3]

The MILL family, composed of MILL1 and MILL2, is a group of nonclassical MHC class I molecules that occur in some orders of mammals. It has been reported that mouse MILL2 is involved in wound healing; however, the molecular mechanisms remain unknown. Here, we determine the crystal structure of MILL2 at 2.15 Å resolution, revealing an organization similar to classical MHC class I. However, the α1-α2 domains are not tightly fixed on the α3-$\beta_2$m domains, indicating unusual interdomain flexibility. The groove between the two helices in the α1-α2 domains is too narrow to permit ligand binding. Notably, an unusual basic patch on the α3 domain is involved in the binding to heparan sulfate which is essential for MILL2 interactions with fibroblasts. These findings suggest that MILL2 has a unique structural architecture and physiological role, with binding to heparan sulfate proteoglycans on fibroblasts possibly regulating cellular recruitment in biological events.

[1] Laboratory of Microbiology, Showa Pharmaceutical University, Machida, Tokyo 190-8543, Japan. [2] Medical Institute of Bioregulation, Kyushu University, Fukuoka 812-8582, Japan. [3] Laboratory of Biomolecular Science, Faculty of Pharmaceutical Sciences, Hokkaido University, Sapporo 060-0812, Japan. [4] Department of Integrated Biosciences, Graduate School of Frontier Sciences, The University of Tokyo, Kashiwa, Chiba 277-8562, Japan. [5] Photon Factory, High Energy Accelerator Research Organization, Tsukuba, Ibaraki 305-0801, Japan. [6] Cardiff Institute of Infection & Immunity, University of Cardiff, Henry Wellcome Building, Heath Park, Cardiff CF14 4XN, UK. [7] Department of Pathology, Faculty of Medicine and Graduate School of Medicine, Hokkaido University, Sapporo 060-8638, Japan. Correspondence and requests for materials should be addressed to K.M. (email: maenaka@pharm.hokudai.ac.jp)

Classical major histocompatibility complex class I (MHC-I) molecules (e.g. human HLA-A, B, C, mouse H-2K, D, L) are highly polymorphic glycoproteins associated with $\beta_2$-microglobulin ($\beta_2$m), expressed on the cell surface of nucleated cells. The main function of classical MHC-I molecules is the cell-surface presentation of peptides, derived from the degradation of cytosolic proteins, to CD8[+] cytotoxic T cells[1]. The extracellular region of MHC-I heavy chains comprises α1, α2, and α3 domains[2]. The membrane-distal α1-α2 domains form two α-helices bordering an anti-parallel β-sheet platform[2]. Cytosolic peptides bind in the groove formed between these helices[2]. T cell receptors on CD8[+] cytotoxic T cells recognize cognate peptide-MHC complexes at the cell surface[3]. During CD8[+] cytotoxic T cell activation, the CD8 co-receptor can enhance interactions with classical MHC-I by binding to the α3 domain[4,5]. If the bound peptide is derived from a foreign antigen or abnormal self-protein, CD8[+] cytotoxic T cells are activated to eliminate the cells[1]. Hence, classical MHC-I molecules are key proteins in the adaptive immune system in vertebrates. The high degree of polymorphic variation in amino acids in the peptide-binding groove increases the repertoire of peptides bound by MHC-I at the population level.

Mammalian genomes also incorporate MHC-I-like nonclassical MHC-I genes[6]. Overall the three-dimensional structures of nonclassical MHC-I molecules resemble classical MHC-I[7]. Although some nonclassical MHC-I molecules display peptides in grooves formed by their α1-α2 domains, these molecules also have additional roles which are not just restricted to the peptide presenting function of classical MHC-I to T cells[8]. HLA-E binds peptides derived from the leader sequence of classical MHC-I for cell surface expression and interacts with CD94/NKG2 receptors on NK cells and T cells[9,10]. In addition, other nonclassical MHC-I molecules bind low-molecular-weight non-peptide ligands in the grooves between the α1-α2 domains, which, in many cases, determines their unique function. Microbial vitamin B metabolites bound to MR1 molecules activate mucosal-associated invariant T (MAIT) cells[11,12]. The CD1 family presents glycolipids to αβ T cells or NKT cells[13]. On the other hand, some nonclassical MHC-I molecules do not apparently bind any low-molecular-weight ligands in the α1-α2 groove[14–18]. Thus, whether ligands bind in the groove formed by the α1-α2 domains plays a critical role in determining the function of nonclassical MHC-I.

We identified a nonclassical MHC-I gene family designated as MILL (MHC-I-like located near the leukocyte receptor complex) in the genomes of rodents[19,20], marsupials[21] and odd-toed ungulates[20]. The members of this family, MILL1 and MILL2, are cell surface $\beta_2$m-associated GPI-anchored glycoproteins[22,23]. MILL1 is exclusively expressed in immune-related tissues, such as thymus and skin, whereas MILL2 is found at very low levels on several different tissues[19,20,22]. The expression level of MILLs is independent of the function of transporter associated with antigen processing (TAP)[22], indicating that cytosolic peptides are not required for their assembly at the cell surface. Thus, MILLs are not involved in peptide antigen presentation to CD8[+] cytotoxic T cells. Phylogenetic analysis indicates that the MILL family is most closely related to the human MIC family[19–21,24], which binds to NKG2D[25]. Furthermore, rodents lack *MIC* genes and conversely, humans do not have *Mill* genes[19–21,24]. This observation suggested that the MILL family, acting as the counterpart for the human MIC family in rodents, might activate NK cells through binding to NKG2D. However, mouse MILL proteins were subsequently shown not to bind to NKG2D[23]. Instead of typical immunological roles, MILLs were reported to be involved in nutrient metabolism and wound healing in mice[23]. These findings suggest that MILL molecules have distinct functions

from classical MHC-I and MIC molecules; however, the physiological roles of MILLs remain largely unknown.

In this study, we present the crystal structure of mouse MILL2 at 2.15 Å resolution. Surprisingly, two conformations of MILL2 with a large difference in domain orientation are observed. One conformation is essentially similar to typical MHC-I molecules, whereas, in the other, the α1-α2 domains are located distantly from the α3-$\beta_2$m domains exposing the interface between the α1-α2 and α3-$\beta_2$m domains. This structural diversity indicates that the α1-α2 domains of MILL2 can associate with $\beta_2$m plastically. On the other hand, the tight proximity of the two helices formed by the α1-α2 domains leaves too little space to bind ligand. Based on these observations, we speculate that MILL2 presumably plays no role in ligand presentation at the cell surface. Interestingly, we find a remarkable basic patch on the β-strand of α3 domain, which is a unique feature of MILL2 compared with other MHC-I. Based on the MILL2 tetramer-staining assay, we show that this basic patch in the α3 domain forms a binding site for heparan sulfate (HS) on the surface of NIH-3T3 cells, indicating that heparan sulfate proteoglycans (HSPGs) on fibroblasts may be physiological ligands for MILL2.

## Results

**Structural determination of mouse MILL2.** Previous study has shown that, although bacterially expressed recombinant MILL2 refolds in isolation, $\beta_2$m facilitates refolding efficiency[22]. The extracellular domain of MILL2 was therefore reconstituted with $\beta_2$m. The heterodimer complex of refolded MILL2 with $\beta_2$m was subjected to size-exclusion chromatography (Supplementary Fig. 1a), followed by cation-exchange chromatography (Supplementary Fig. 1b). Highly purified MILL2 that had a 1:1 stoichiometry with $\beta_2$m (Supplementary Fig. 1c) was used for crystallization screening. Crystals of MILL2 were obtained in buffer containing polyethylene glycol 3350 and sodium sulfate (Supplementary Fig. 1d). X-ray diffraction data were collected up to 2.15 Å. The structure was solved by the molecular replacement method using the crystal structure of human hemochromatosis protein (PDB ID: 1A6Z) as a search model. Supplementary Table 1 presents a summary of the statistics for structural refinement. A section of the 2*Fo-Fc* electron density map is shown in Supplementary Fig. 1e.

**MILL2 can adopt two distinct conformations.** Unexpectedly, two different conformations of MILL2 were observed in the asymmetric crystal unit (Supplementary Fig. 2). The overall structure of one conformation closely resembled other typical MHC-I molecules[2] harboring three domains (α1, α2, and α3) with $\beta_2$m (Fig. 1a and Supplementary Fig. 2, chains in purple (MILL2) and green ($\beta_2$m)). The α1 and α2 domains, composed of residues 6–179, have two anti-parallel α-helices lying on an anti-parallel β-sheet platform formed by seven β-strands (Supplementary Fig. 3). In ligand-presenting MHC-I molecules, small molecules bind in the groove formed between these α1 and α2 helices[7]. The α3 domain, composed of residues 188–275, is annotated as an immunoglobulin-like C1 domain[19] (Supplementary Fig. 3). Similar to other $\beta_2$m-associated MHC-I, $\beta_2$m was located to the side of this domain (Supplementary Fig. 3). Five N-terminal MILL-specific residues and the loop formed by residues 179–183 were not observed in this structure, suggesting that they are disordered.

In the second conformation, each domain exhibits a structure similar to the typical MHC-I domains; however, the α1-α2 domains do not associate with $\beta_2$m (Fig. 1a, Supplementary Fig. 2, 3, chains in magenta (MILL2) and yellow ($\beta_2$m)). Superimposition of the typical closed conformation and the unique open

conformation clearly shows that the α1-α2 domains are located far from the α3-β2m domains in the open conformation (Fig. 1a). MILL2 is the first β2m-associated MHC-I molecule shown to form both open and closed conformations. To date, all β2m-associated MHC-I molecules have been reported to form the closed conformation with the α1-α2 domains typically fixed onto the α3-β2m domains by tight contact with β2m (Fig. 1b, left). On the other hand, the α1-α2 domains of β2m-free MHC-I molecules (e.g. human MICA and MICB) form open conformations because the α1-α2 domains are no longer anchored in the absence of β2m, (Fig. 1b, right)[14,26]. The existence of both open and closed conformations for MILL2 indicates that, although the α1-α2 domains associate with β2m, this interaction is not of sufficiently high avidity to maintain a constitutive association (Fig. 1b, center).

Small angle X-ray scattering (SAXS) profiles of MILL2 were compared with theoretically calculated scattering curves of both the open and the closed conformation from the crystal structures (Supplementary Fig. 4). It shows good agreement with the closed conformation with the chi-square value of 0.989. The SAXS experimental conditions and the resultant values of analysis are summarized in Supplementary Table 2.

A similar number of residues are involved in contacts between MILL2 and β2m compared to those formed by classical mouse MHC-I, H-2D$^b$ (PDB ID: 1CE6)[27] (Supplementary Data 1). Thirty contact residues are found between H-2D$^b$ and β2m in the H-2D$^b$/β2m complex, whereas 30 residues form contacts between MILL2 and β2m in the closed conformation of MILL2 (Supplementary Data 1). The binding surfaces with β2m are very similar between MILL2 and H-2D$^b$. Contact residues on β2m are also well conserved; however, those on the heavy chains are not (Fig. 1c, d and Supplementary Fig. 5a, b). In particular, on the α1-α2 domains, only three of a total of 30 contact residues are conserved between H-2D$^b$ and MILL2 (Supplementary Data 1). In addition, there are significantly fewer van der Waals interactions in the MILL2/β2m complex (Supplementary Data 1). Whereas H-2D$^b$ forms 201 van der Waals interactions with β2m, MILL2 forms only 165 interactions (Supplementary Data 1). In particular, the α1-α2 domains of H-2D$^b$ form 108 interactions but the corresponding domains of MILL2 form only 97 (Supplementary Data 1). These results demonstrate that although MILL2 associates with β2m by the same interface, the molecular packing between MILL2 and β2m is less tight than that of classical MHC-I.

In H-2D$^b$, the main contact sites on the α3 domain are concentrated in a region between Arg234 and Gln242[28]. In the corresponding region of MILL2, contacts with Pro238, Gly240, Asp241 and Gln245 were conserved (Supplementary Data 1). Multiple van der Waals interactions and hydrogen bonds with β2m were observed in this region of MILL2 (Supplementary Data 1), suggesting that the interactions of the α3 domain with β2m are sufficiently strong for anchoring β2m without the need for additional interactions with the α1-α2 domains.

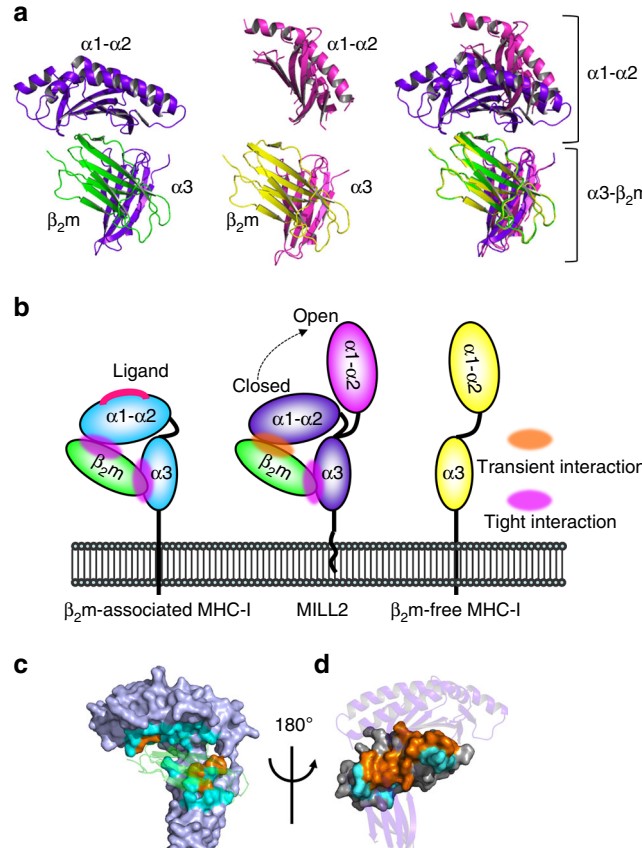

**Fig. 1** MILL2 crystals form both closed and open conformations. **a** The closed conformation (left), open conformation (center), and superimposition of these two conformations based on the position of β2m chain (right) are represented by ribbon diagrams. Purple, closed conformation of MILL2; magenta, open conformation of MILL2; green, β2m associated with the closed conformation of MILL2; yellow, β2m associated with the open conformation of MILL2. **b** Schematic model of the flexible membrane-distal α1-α2 domains (center). In typical β2m-associated MHC-I molecules, the α1-α2 domains have no flexibility because of tight interactions with β2m (left). By contrast, the α1-α2 domains of β2m–free MHC-I molecules (e.g. human MIC family) are located far from the membrane-proximal α3 domain. **c** Surface model of MILL2 showing contact residues with β2m. Orange, conserved contact residues with β2m on MILL2 and H-2D$^b$; cyan, MILL2-specific contact residues with β2m. β2m is represented by a transparent ribbon diagram (green). **d** Surface model showing contact residues on β2m with MILL2. Orange, contact residues on β2m conserved between MILL2 and H-2D$^b$; cyan, MILL2-specific contact residues on β2m. MILL2 heavy chains are represented by a ribbon diagram (purple)

**Narrow spacing of the two helices in the α1-α2 domains**. The top view of MILL2 is shown in Fig. 2. The two helices in the α1-α2 domains are very close to each other (Fig. 2a). The distance between the two helices of the α1-α2 domains is much smaller than that of classical MHC-I (PDB ID: 1E27) (Fig. 2b). The space between the two helices in classical MHC-I is large enough to permit binding of cytosolic peptides (Fig. 2e). By contrast, the interhelical space of MILL2 is filled with amino acid side chains from both helices (Fig. 2d). Such narrow interhelical spacing is also observed for NKG2D ligand members, which do not present ligands for their function, e.g. human MICB (PDB ID: 1JE6)[26] (Fig. 2f). Notably, the space between the α1-α2 helices is

narrower than the corresponding space in MICB (Fig. 2c). Thus, it is unlikely that ligands are bound in the groove formed between the two helices of the α1-α2 domains in MILL2, and the physiological function of MILL2 is unlikely to be related to ligand presentation.

**Two remarkable basic patches on MILL2**. Electrostatic analysis revealed that both conformations of MILL2 have a basic patch between the two helices of the α1-α2 domains (Supplementary Fig. 6a, b). This patch is formed by Arg65, Lys72, Lys76, His169, and Arg172 of MILL2 (Supplementary Fig. 7a). The amino acid sequences of MILLs are most closely related to human MICA and

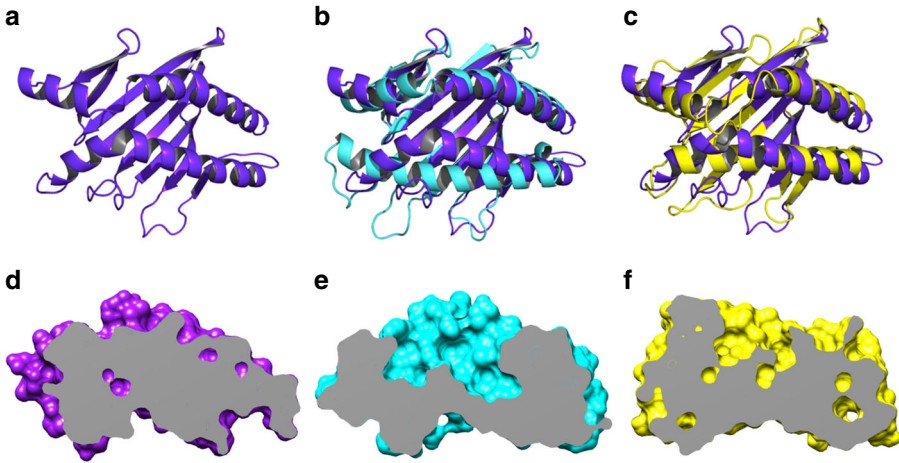

**Fig. 2** The narrow space between the two helices in α1-α2 domains of MILL2. **a** Top view of the membrane-distal α1-α2 domains of MILL2 represented by a ribbon diagram (purple). **b**, **c** Superimposition of MILL2 α1-α2 domains with **b** HLA-B (PDB ID: 1E27) or **c** MICB (PDB ID: 1JE6), represented by ribbon diagrams. Purple, MILL2; cyan, HLA-B; yellow, MICB. **d–f** Side view of the α1-α2 domains of **d** MILL2, **e** HLA-B (PDB ID: 1E27), and **f** MICB (PDB ID: 1JE6) represented by cutaway model. Purple, MILL2; cyan, HLA-B; yellow, MICB

MICB[19] which activate NK cells by binding to the NKG2D receptor. However, human MIC and other NKG2D family ligands have an acidic patch on the top of the α1-α2 domains[29,30] (Supplementary Fig. 8). Electrostatic interactions between NKG2D and its ligands are important for NK cell activation[29,30]. Thus, the presence of a basic patch in MILLs in place of the characteristic acidic patch of NKG2D ligands provides a plausible explanation for their lack of interaction with this receptor[23].

Interestingly, electrostatic analysis also revealed another large basic patch located on the exposed surface of the α3 domain in both closed and open conformations (Fig. 3a, b). This region is composed of Arg194, Arg200, Lys229, Arg232, Arg247 and Arg251 (Supplementary Fig. 7b). Because these basic patches are unique to MILL2 among all MHC-I superfamily members, we hypothesized that these two remarkable basic patches are involved in the physiological functions of MILL2. The physiological receptor(s) of MILL2 remain(s) unknown. However, recombinant MILL2 Fc-fusion protein (MILL2-Fc) binds to the cell surface of the mouse NIH-3T3 fibroblast cell line[23]. Our MILL2 tetramer also bound to this cell line (Supplementary Fig. 9) . Trypsin treatment of the cell surface completely abolished tetramer binding (Supplementary Fig. 9), suggesting that NIH-3T3 cells express a putative cell surface receptor for MILL2. In order to determine the possible involvement of the basic patches on the molecule in binding to NIH-3T3 cells, we generated MILL2-tetramers with alanine-substitutions of the arginine and lysine residues forming these regions. Substitutions of K72A + K76A and R65A + R172A were selected to change the basic patch in the α1-α2 domains, whereas substitutions of R194A + R200A + R251A and K229A + R232A + R247A were generated to change the basic patch in the α3 domain. K72A + K76A and R65A + R172A mutant tetramers bound to the cell surface of NIH-3T3 cells similar to wild type MILL2 tetramer (Fig. 3c). By contrast, R194A + R200A + R251A and K229A + R232A + R247A mutagenesis completely abrogated tetramer binding to NIH-3T3 cells (Fig. 3c). The structural stability of these mutants was confirmed by size exclusion chromatography (Supplementary Fig. 10). While Arg247 binds to β2m, K229A + R232A + R247A mutagenesis did not lose the association with β2m (Supplementary Fig. 10). Hence, these results suggest that the unusual basic patch in the α3 domain is a potential binding site for the putative receptor on NIH-3T3 cells.

**MILL2 binds to heparan sulfate proteoglycans**. In the crystal structure of MILL2, a SO4 ion from the crystallization buffer is located on the edge of the unique basic patch in the α3 domain, proximal to Arg251 (Fig. 3d). Highly basic patches on protein surfaces can bind to the HS moiety of HSPGs, which are negatively charged glycosaminoglycans[31–33]. We reasoned that this observation gave some indication of the physiological ligand for MILL2 and hypothesized that MILL2 could bind to HS through the basic patch located on the surface of the α3 domain. Consistent with MILL2 binding to HS, MILL2 tetramers did not bind to NIH-3T3 cells after treatment with the HS-specific endoglycosidases heparinase I and III (Fig. 3e). In addition, soluble heparin blocked MILL2 tetramer binding to NIH-3T3 cells (Fig. 3f). Furthermore, cell surface trypsinization of NIH-3T3 cells completely abolished tetramer binding (Supplementary Fig. 9). These results suggest that MILL2 binds to cell surface HSPGs on fibroblast cells.

We performed heparin affinity chromatography to determine whether MILL2 interacts directly with heparan sulfate. Wild type MILL2 bound to heparin agarose and was eluted with 410 mM NaCl (Fig. 4a, b). Interestingly, although the heavy chain bound to heparin agarose, β2m dissociated from the heavy chain and eluted in the flow-through fraction (Fig. 4b). On the other hand, the R194A + R200A + R251A mutant, whose tetramer does not bind to the surface of NIH-3T3 cells (Fig. 3c), hardly bound to the column, and mainly eluted in the flow-through fraction (Fig. 4a, b). Any small quantities of bound mutant MILL2 eluted at the lower concentration of NaCl (130 mM) compared to wild type MILL2 (Fig. 4a and b). We next studied interaction of an MHC-I molecule, which lacks the basic patch found in MILL2 (HLA-G, Supplementary Fig. 11) to heparin agarose. HLA-G did not bind to the column (Fig. 4a, b). These results strongly support that MILL2 directly and specifically binds to heparan sulfate at the basic patch in the α3 domain.

**Discussion**

In this study, we determined the three-dimensional structure of mouse MILL2 by X-ray crystallography. The domain structures of MILL2 are similar to those of MHC-I molecules which typically associate with β2m. However, distinct structural features shed light on interesting and unique functions of MILL2. Notably, an

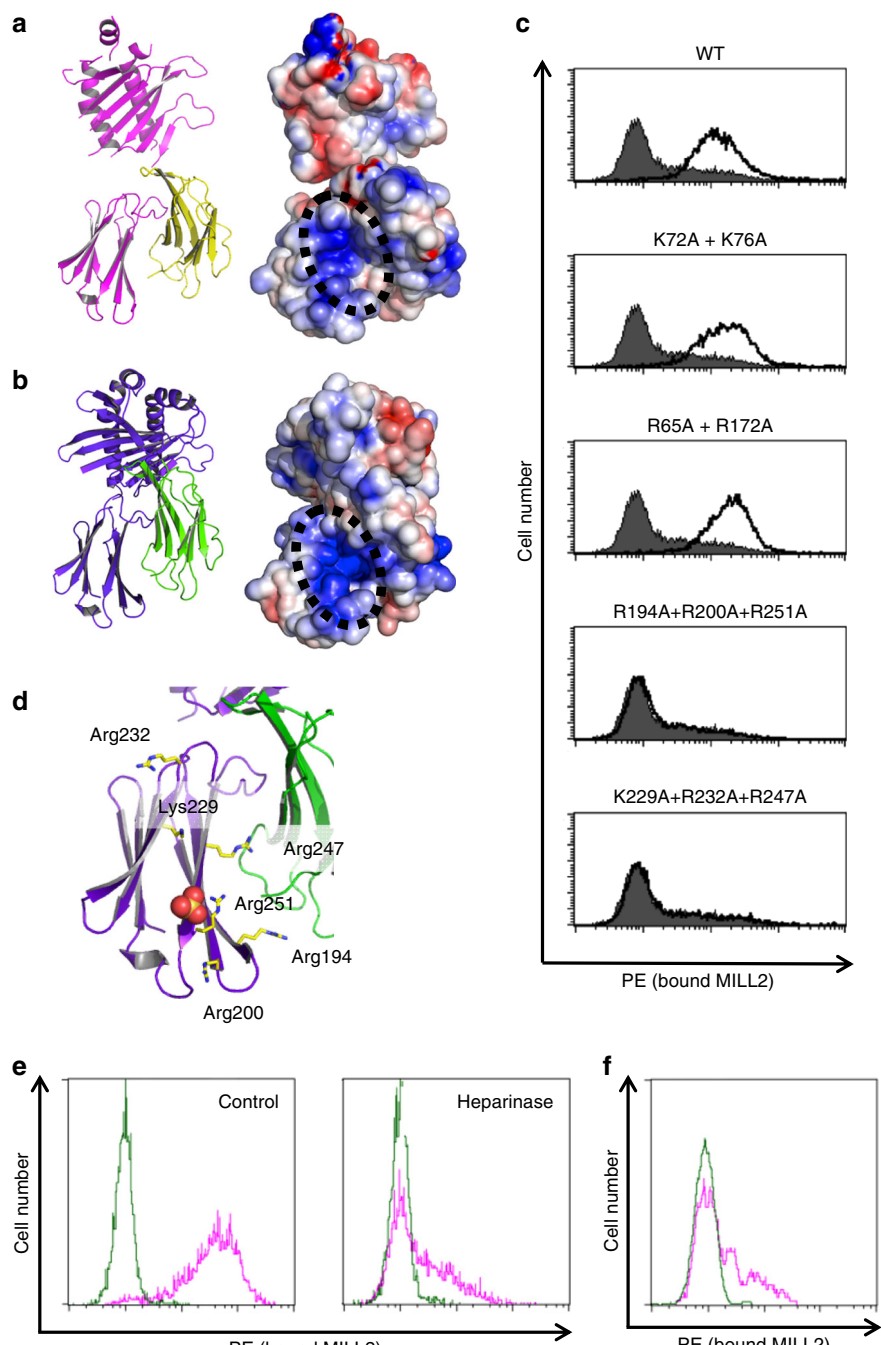

**Fig. 3** Putative heparan sulfate-binding site on the α3 domain of MILL2. **a**, **b** Side views of **a** the open and **b** the closed conformation of MILL2 shown by a ribbon diagram (left) and its electrostatic surface potential model (right). Magenta and yellow indicate open conformation of MILL2 and β2m, respectively. Purple and green indicate closed conformation of MILL2 and β2m, respectively. Red and blue indicate negatively and positively charged areas, respectively. The black dotted circles highlight the position of the basic patch. **c** Histograms show the binding levels of MILL2 and the respective alanine-substituted tetramers to NIH-3T3 cells. PE-conjugated MILL2 tetramer and tetramers incorporating alanine substitutions in the basic patch of the α1-α2 domains (K72 + K76A and R65A + K172A) or α3 domain (R194A + R200A + R251A, K229A + R232A + R247A) were generated. Staining with PE-conjugated streptavidin (shaded histograms) or PE-conjugated tetramer (open histograms) was measured by flow cytometry. **d** An SO4 ion is located near the basic patch on the α3 domain in MILL2 crystals. The α3 domain of MILL2 is shown as a ribbon diagram (purple) and sticks represent the side chains of residues in this patch. The SO4 ion is shown as a space-filling model. **e** Histograms show staining of NIH-3T3 cells with wild type MILL2 tetramers with or without heparinase treatment. Cells were preincubated with PBS (control) or PBS including heparinase (heparinase). After treatment, cells were stained with PE-conjugated streptavidin (green histogram) or PE-conjugated MILL2 tetramer (pink histogram) and binding was measured by flow cytometry. **f** Histogram shows heparin-competition for the binding of MILL2 tetramers to NIH-3T3 cells. PE-conjugated streptavidin (green histogram) or PE-conjugated MILL2 tetramer (pink histogram) staining was performed with heparin. Binding was measured by flow cytometry

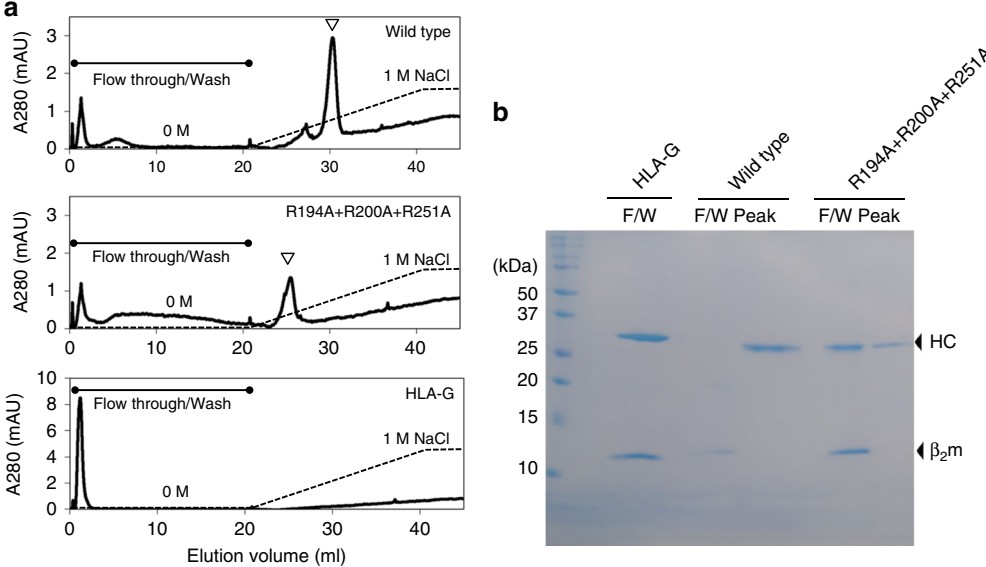

**Fig. 4** MILL2 binds directly to heparin. **a** Heparin affinity chromatography. MILL2 (Wild type or R194A + R200A + R251A) or HLA-G was loaded on a HiTrap Heparin HP column equilibrated with 20 mM Tris-HCl buffer pH 8.0. Subsequently, column were washed with 20 column volumes of 20 mM Tris-HCl buffer pH 8.0 and bound proteins were eluted with 20 column volumes of a linear NaCl gradient (0 to 1 M NaCl in 20 mM Tris-HCl buffer pH 8.0). Open arrowheads indicate peaks eluted by the NaCl gradient. **b** SDS-PAGE analysis of collected flow-through/wash fraction (F/W) and eluted peak fraction (Peak). Arrows indicate heavy chain (HC) or $\beta_2$m. An uncropped scanned image of this gel is shown in Supplementary Fig. 15

unusual basic patch on the α3 domain of MILL2 bestows HS binding capacity.

Firstly, our structural analysis indicates MILL2 associates with $\beta_2$m; however, the interaction between the α1-α2 domains of MILL2 and $\beta_2$m is not as strong as other $\beta_2$m-associated MHC-I (Fig. 1a). $\beta_2$m-associated MHC-I molecules are not stably expressed at the cell surface in the absence of $\beta_2$m[34]. Among the MHC-I superfamily, AZGP1, MICA, MICB and other NKG2D ligands form $\beta_2$m-free MHC-I molecules, which do not require $\beta_2$m for their correct folding and function. Some NKG2D ligand family members (i.e. Rae-1, H60 and ULBP) even lack an α3 domain. In the case of MILL2, although $\beta_2$m associates with endogenous MILL2 and increases refolding efficiency, it is not essential for refolding in vitro[22]. Moreover, MILL2 is expressed on the cell surface of $\beta_2$m-deficient cell lines[23]. These observations indicate that $\beta_2$m is not required for the structural stability of MILL2. In this study, MILL2 adopted two alternative conformations in crystals (Fig. 1a and Supplementary Fig. 2). The presence of an open conformation suggested that, although $\beta_2$m does associate with the α1-α2 domains of MILL2, this structure is not as fixed as other $\beta_2$m-associated MHC-I (Fig. 1a). The residues at the interface between MHC-I molecules and $\beta_2$m are highly conserved among all MHC superfamily members which associate with $\beta_2$m[28]. However, many of these residues are not conserved in the MILL2 heavy chain (Fig. 1c). Moreover, the number of physical interactions with $\beta_2$m is also significantly fewer than those seen with classical MHC-I (Supplementary Data 1). These observations suggest that $\beta_2$m supports MILL2 rather than forming an essential element of its structure. SAXS analysis shows that the majority of MILL2 in solution exists in the closed conformation (Supplementary Fig. 4). The closed conformation may be advantageous for keeping MILL2 stability in solution. However, the crystal packing definitely indicates the presence of the open conformation is not the result of interference of the neighboring molecules (Supplementary Fig. 2). Interestingly, binding to heparin by the basic patch on MILL2 released $\beta_2$m (Fig. 4), indicating that heparin-bound MILL2

adopts the open conformation. The close contacts required for strong binding to heparan sulfate may make it necessary to change the conformation from closed to open by releasing $\beta_2$m. Therefore, our findings suggest that the conformational flexibility observed here is physiological and further elucidation of its functional significance will be required in future work.

Secondly, the space between the two helices in the α1-α2 domains was too narrow to permit binding of peptides or small ligands which typically associate with MHC-I (Fig. 2d). MHC-I molecules which bind peptide generally do not refold in the absence of peptide, because peptide-binding is believed to stabilize the structure of the α1-α2 domains of MHC-I[35]. We have previously demonstrated that MILL2 successfully refolds without any peptides or other small ligands[22]. In this study, MILL2 was successfully crystallized in the absence of small molecule ligands. We therefore speculate that, similar to the human MIC family, MILL2 does not bind peptides or small ligands in the groove formed by the α1-α2 domains.

Thirdly, we report the identification of an important basic patch on the α3 domain which mediates binding to HS on the cell surface of fibroblast cell lines. Electrostatic analysis of the crystal structure of MILL2 identified two unique basic patches at the surface of the α1-α2 and α3 domains (Fig. 3a, b and Supplementary Figs. 6, 7). Alanine-substitution of lysine or arginine residues in these patches revealed that the basic patch on the α3 domain is essential for MILL2 tetramer binding to NIH-3T3 cells (Fig. 3c). In addition, heparinase and protease treatment revealed that MILL2 tetramers bind to HSPGs on the surface of NIH-3T3 cells (Fig. 3e, f and Supplementary Fig. 9). Notably, a $SO_4$ ion from the crystallization buffer bound to the basic patch on the α3 domain in crystals (Fig. 3d). Taken together, these findings support that the charge–charge interactions between the basic patch on the α3 domain and the HS moiety of HSPGs on fibroblast cells mediate binding. In this study we report an MHC family member with a unique basic patch on the α3 domain (Supplementary Fig. 11). In both the open and closed conformations, the basic patch on the α3 domain surface is mainly composed of the fourth

and fifth β-strand of this domain (Supplementary Fig. 12a). Lys229 and Arg232 are located on the fourth β-strand, and Arg247 and Arg251 are on the fifth β-strand (Supplementary Fig. 12a). Interestingly, classical MHC-I molecules have a long loop (known as the CD loop) instead of a fourth β-strand (Supplementary Fig. 12c). The CD loop is the most important region for association with the CD8 co-receptor[36,37]. By contrast with MILL2, the region corresponding to the fifth β-strand in classical MHC-I molecules is buried beneath the CD loop (Supplementary Fig. 12b). However, the fifth β-strand in the MILL2 α3 domain is exposed to solvent because the location of the fourth strand makes the fifth strand accessible (Supplementary Fig. 12b). All reported MHC-I structures have the classical MHC-I type CD loop, supporting that the basic patch on this MILL2-specific strand is a unique structure for specific binding of HS. To date, CD8[5,36,37], LILRs[38], PIRs[39] and Ly49s[40–42] are also known as the α3-associating MHC receptors; however, receptor binding to basic patches on MHC heavy chains is a unique mode of interaction. The binding of MILL2 to HS by a basic patch on the α3 domain therefore constitutes a yet another binding mode for MHC-I family member receptors which has not been previously reported.

It was previously reported that MILL2 could be involved in nutrient metabolism[23]. MILL2 binding to bone marrow cells was reduced by the presence of some polyamines (i.e., spermine and spermidine) or fetal bovine serum (FBS)[23]. This phenomenon suggests that effects of MILL2 on nutrient metabolism are modulated by binding to serum factors[23]. The present finding, namely the binding of MILL2 to HSPGs, suggests that positively charged polyamines or serum factors may competitively inhibit the interaction between the basic patch on the α3 domain and complementary negatively charged HS.

The physiological role and cognate receptor(s) for the other MILL family member, MILL1, are unknown. The amino acid identity of the extracellular domains of MILL1 and MILL2 is 70%[19]. Our previous report revealed that whereas β2m was not required for the folding of MILL2, it was essential for correct folding of MILL1[22]. Moreover, endogenous MILL1 is not expressed on thymic stromal cells from β2m-deficient mice[22]. These findings indicate that association with β2m is essential to stabilize the structure of MILL1. Amino acid sequence comparisons indicate that residues in the α2 and α3 domains which bind β2m are well conserved, whereas many of the residues that bind β2m in the α1 domain differ between MILL1 and MILL2 (Supplementary Fig. 13). It is possible that strong interactions in the α1 domain of MILL1 with β2m stabilize its structure. On the other hand, MILL1 lacks several amino acid residues that form the basic patch on the MILL2 α3 domain (Supplementary Fig. 14). Our tetramer binding assays revealed that the wild type MILL1 tetramer does not bind to NIH-3T3 cells (Supplementary Fig. 9). Although previous reports demonstrated that MILL1-Fc fusion protein could bind to NIH-3T3 cells, the degree of binding was much lower than for MILL2-Fc fusion protein[23]. These results suggest that cell surface HSPGs on NIH-3T3 cells are not physiological ligands for MILL1.

In this study, we successfully identified HS as a ligand for MILL2, shedding light on the physiological role of MILL2. Many types of HSPGs are expressed on fibroblasts, epithelial cells and as part of the extracellular matrix[31,32]. The interactions between HSPGs and their ligands or receptors are involved in multiple biological pathways in vivo (e.g., cell growth, inflammation and wound healing)[31,32]. Indeed, Ravinovich et al. demonstrated that administration of a MILL2-specific monoclonal antibody enhances wound healing in the skin[23]. The syndecan family of HSPGs are key molecules in wound healing[43–46]. Thus, it is plausible that MILL2 binding to HS on syndecans could play an important role in the recruitment of MILL2-expressing cells to fibroblasts by utilizing HS and HSPGs, in a way that is similar to many other signaling systems, such as TGF-β, FGF and Wnt. Identification of HSPGs associated with MILL2 and investigation of the role of their interaction in controlling the mobility or proliferation of fibroblasts are likely to lead to a further understanding of the physiological functions of MILL2.

## Methods

**Expression of recombinant MILL2 and β2m.** The extracellular domains of mouse MILL2 and mouse β2m were expressed individually in *E. coli* strain BL21 (DE3) pLysS (Agilent) using the pGMT7 expression vector[47]. DNA fragments were obtained by PCR using the BALB/c mouse cDNAs as templates[19]. The primer sequences that incorporated *Psh*BI and *Hin*dIII sites to facilitate cloning are shown in Supplementary Table 3. The forward primer for MILL2 introduced synonymous codon changes destabilizing secondary structure formation by mRNA. PCR products were digested with *Psh*BI/*Hin*dIII and ligated to the *Nde*I/*Hin*dIII-digested pGMT7 vector (hereafter designated as pGMT7-MILL2 and pGMT7-β2m). In all cases, the integrity of expression constructs was verified by sequencing analysis. Plasmid DNAs for transformation were isolated with the plasmid purification kit purchased from QIAGEN. To express recombinant MILL2 and β2m, pGMT7-MILL2 or pGMT7-β2m were individually transformed into *E. coli* strain, BL21 (DE3) pLysS. Recombinant proteins were expressed as inclusion bodies by adding isopropyl-1-thio-β-D-galactopyranoside. Five hours after induction, cells were harvested and lysed by sonication in suspension buffer containing 1% Triton X-100, 50 mM Tris-HCl (pH 8.0) and 150 mM NaCl. Inclusion bodies of overexpressed protein were obtained by centrifugation at $7000 \times g$ for 10 min at 4 °C. Isolated inclusion bodies were denatured by solubilization buffer containing 50 mM Tris-HCl (pH 8.0), 6 M guanidine hydrochloride, 100 mM NaCl and 10 mM EDTA. Denatured inclusion bodies were stored at −80 °C until use.

**Refolding and purification of soluble MILL2/β2m heterodimer.** Soluble MILL2/β2m complexes were prepared by rapid dilution refolding. Firstly, denatured inclusion bodies of β2m with 10 mM DTT were diluted in refolding buffer (0.1 M Tris-HCl (pH 8.0), 0.4 M L-arginine-HCl, 6.5 mM cysteamine, and 3.7 mM cystamine) to a final protein concentration of 10 µM with gently stirring for 48 h at 4 °C. Successfully refolded β2m was purified by size-exclusion chromatography with HiLoad 26/60 Superdex™ 75 prep grade (GE Healthcare). Subsequently, denatured inclusion bodies of MILL2 with 10 mM DTT were diluted in refolding buffer with 10 µM refolded β2m to a final MILL2 protein concentration of 1 µM with stirring for 48 h at 4 °C. Successfully refolded complexes of MILL2/β2m were purified by size-exclusion column chromatography using a HiLoad 26/60 Superdex™ 75 prep grade column and then cation-exchange column chromatography using RESOURCE S (GE Healthcare). The fraction corresponding to the MILL2/β2m complex was collected and concentrated to 9.5 mg ml⁻¹ in 10 mM HEPES-NaOH (pH 7.0) for crystallization.

**Crystallization and structural determination.** Crystals of MILL2 were grown at 20 °C by the hanging drop vapor-diffusion method. The final crystallization condition was 0.1 M BIS-Tris propane·HCl (pH 8.5), 20% (w/v) polyethylene glycol 3350, and 0.2 M sodium sulfate. For data collection, crystals were soaked in crystallization buffer supplemented with 25% ethylene glycerol, and then flash frozen in a cryostream at 100 K. The final data set was collected at beamline BL17A of Photon Factory (Tsukuba, Japan) using an ADSC CCD detector Q270. The diffraction data set was integrated, merged, and scaled with the HKL2000 program package[48]. The structure was solved by the molecular replacement method using the Molrep program[49]. Structure refinement was carried out using Phenix[50]. The stereochemical properties of the structure were assessed using Molprobity[51] and COOT[52]. Figures of structural information were prepared with PyMOL 1.4 (http://pymol.sourceforge.net). Intermolecular contact atoms were identified using CONTACT in CCP4i[53] and electrostatic surface potentials were calculated using APBS[54]. Cutaway models were prepared with UCSF Chimera[55].

**SAXS data collection and processing.** SEC-SAXS data were obtained at the BL-10C, Photon Factory (Tsukuba, Japan). An Acquity HPLC system of UPLC H-Class system (Waters) with the Superdex 200 Increase 10/300 GL (GE Healthcare) was utilized to isolate MILL2. The column was preequilibrated with 10 mM HEPES-NaOH, 150 mM NaCl, pH 7.4 buffer. The flow rate during the sample measurement was set at 0.1 ml min⁻¹. The eluted sample was loaded into a stainless-steel cell with a H1.5 × W3.0 × T1.0 mm window size and a 0.02 mm-thick quartz glass window and exposed to X-ray and UV-Visible light simultaneously in order to evaluate the sample concentration at the X-ray exposed position correctly. The scattering intensities ($I(Q)$) were measured in the region of $0.00753 < Q < 0.424$ Å⁻¹, where $Q = 4\pi \sin\theta/\lambda$ and $\lambda = 1$ Å, at a distance between sample and detector of 2.009 m. The exposure times and the number of images were 20 s and 296 images, respectively. These images were recorded on a

PILATUS3 2 M detector (Dectris). Before injection of the sample on the column, 15 images of flowed buffer scattering were collected for background data. The scattering intensities recorded on the 2D image were azimuthally averaged to convert the 1D profile and were subtracted from the background intensities by using SAngler[56]. The conversion of intensities to an absolute scale was performed by using water scattering as a standard. Since no concentration dependence was observed for the whole data, the five data around the peak derived from MILL2 were averaged as final scattering data. The scattering profiles obtained from X-ray crystal structures were calculated using CRYSOL[57]. The values of the scattering intensity at zero angle ($I(0)$) and the radius of gyration ($R_g$) were obtained by Guinier analysis ($Q \times R_g < 1.3$) by using AUTORG[58].

**Tetramer binding assay.** cDNAs for MILL1 and MILL2 mutants without stop codon were generated by PCR. The primer sequences that incorporated *Psh*BI and *Bam*HI sites to facilitate cloning are shown in Supplementary Table 3. The PCR products were digested with *Psh*BI/*Bam*HI and ligated to the *Nde*I/*Bam*HI-digested pGMT7-birA vector[59]. Recombinant soluble MILL1-birA/β2m and MILL2-birA/β2m heterodimers incorporating the C-terminal biotin ligase (BirA) recognition sequence (GSLHHILDAQKMVWNHR) were prepared by the same procedure mentioned above. Refolded MILL1-birA/β2m and MILL2-birA/β2m were biotinylated with BirA enzyme (Avidity). After biotinylation, biotinylated MILLs were separated from free biotin by size-exclusion chromatography with Superdex™ 200 10/300 GL (GE Healthcare). MILL tetramer was produced by incubating biotinylated MILL/β2m with phycoerythrin (PE)-conjugated strep-tavidin (Thermo) for 15 min on ice. For surface staining with MILL tetramer, NIH-3T3 cells (RCB0150, RIKEN Cell Bank) were detached by DPBS(-) supplemented with 5 mM EDTA, and cell suspensions were stained with PE-conjugated MILL tetramer on ice for 15 min. Stained cells were washed with DPBS(-) supplemented with 2% FBS and binding of MILL tetramers was analyzed by flow cytometry.

**Heparinase and heparin treatment.** For cell surface heparinase treatment, NIH-3T3 cells were detached by DPBS(-) supplemented with 5 mM EDTA, and treated with 2 U ml$^{-1}$ heparinase I and III (Sigma-Aldrich) in DPBS(-) for 60 min at 37 °C. Cells were then stained with tetramers after washing twice with ice-cold DPBS(-). For studying effects of heparin on MILL2 binding, detached cells were stained with MILL2 tetramer in the presence of 10 μg ml$^{-1}$ heparin (sodium salt, Nacalai tesque).

**Heparin affinity chromatography.** Heparin affinity chromatography was performed with a HiTrap Heparin HP column (1 ml, GE Healthcare) on the AKTA pure system (GE Healthcare). HLA-G protein was overexpressed as inclusion bodies in *E. coli* using the pGMT7 HLA-G vector[38] and refolded with human β2m and RIIPRHLQL peptide following the same procedure for MILL2. MILL2 or HLA-G proteins were loaded on a column equilibrated with 20 mM Tris-HCl (pH 8.0). After washing the column with 20 column volumes of equilibration buffer, bound proteins were eluted with 20 column volumes of a 0 to 1 M NaCl in 20 mM Tris-HCl (pH 8.0) linear concentration gradient. Flow through/wash and eluted peak fractions were pooled, concentrated and analyzed by SDS-PAGE. Gels were stained with CBB-G250 solution.

## Data availability

Coordinates and structure factors have been deposited in the Protein Data Bank under the accession code 6A97. Other data are available from the corresponding author upon reasonable request.

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

## Acknowledgements
We thank M. Ohtsu, I. Ohki, K. Sasaki-Tabata, M. Shiroishi, N. Maita, D. Kohda, S. Niizuma, Y. Watanabe and S. Ishido for helpful discussions. We also thank S. Wakatsuki, R. Kato, N. Igarashi, M. Kawasaki, N. Matsugaki, Y. Yamada, M. Senda and T. Senda for assistance in data collection at Photon Factory. This work was supported in part by Japan Society for the Promotion of Science (JSPS) Grants-in-Aid for Scientific Research KAKENHI (Grants 20057020, and 22121007), JSPS Strategic Young Researcher Overseas Visits Program for Accelerating Brain Circulation, Platform Project for Supporting Drug Discovery and Life Science Research (Basis for Supporting Innovative Drug Discovery and Life Science Research (BINDS)) from AMED under Grant Number 18am0101093j0002, the Platform for Drug Discovery, Informatics, and Structural Life Science and the Ministry of Education, Science, Sports, Culture and Technology and the Ministry of Health, Labor and Welfare of Japan, and the Japan Bio-oriented Technology Research Advancement Institute (BRAIN), Hokkaido University Biosurface project and Takeda Science Foundation.

## Author contributions
M. Kajikawa, M. Kasahara, and K.M. designed research; M. Kajikawa, T.O., Y.F., Y.O., K.Y., N.S. and K.M. performed research; M. Kajikawa, T.O., N.M., S.K., M. Kasahara and K.M. analyzed the data; M. Kajikawa, T.O., S.K. and K.M. wrote the paper.

## Additional information

**Competing interests:** The authors declare no competing interests.

