## [Peer Review File · Nature Communications]

Reviewers' Comments:

Reviewer #1:

Remarks to the Author:

Kajikawa and colleagues present data describing the structure of a newly described MHC Class I-like protein, MILL2. MILL2 is only found in mice, marsupials, and odd-toed ungulates and has no known homologue in humans, but is described as being most similar to the MIC MHC-like family of protein in humans. MILL2 has no known ligands, functions in a peptide and TAP independent manner, and has been suggested to play roles in wound healing and metabolism. However, the physiological roles of the MILL family of proteins is largely unknown. In this manuscript, the authors show that MILL2 forms a complex with $\beta 2m$ but can adopt two separate structures, a closed, classical MHC-like structure and an open conformation. This open conformation is similar to that seen for MICA and MICB. However, unlike these proteins, MILL2 is bound to $\beta 2m$.

Extending from the structure, the authors note the presence of a basic patch on the surface of MILL2 that localizes between both the $\alpha 1$ - $\alpha 2$ and $\alpha 3$ domains. The authors then show in the supplement that this basic patch is unique to MILL2, where other MHC Class I and Class I-like proteins typically maintain a more acidic nature here. From here, the authors then hypothesize and test whether or not MILL2 can bind heparan sulfate (HS) and HSPGs found on the cellular surface of NIH-3T3 cells using cell sorting analyses. These results are then extended, showing that heparinase as well as mutagenesis of this basic patch in MILL2 can abrogate the ability of MILL2 to bind to the cellular surface. As a result, the authors suggest that HSPGs may function as ligands for MILL2 and may play a role in its physiological function and wound healing.

Overall, the paper is significant in that a structure of a newly classified MHC Class I-like molecule is resolved and determines a new interaction of this protein with HS and HSPGs. While these findings are significant and worthy of publication, additional work needs to be performed to push the impact of these findings for publication in Nature Communications. Suggestions for improvements (minor and major) are outlined below.

1. Starting on page 6 and through page 7 (lines 106 through 122), the details of the crystallographic methods is described in some details. This information would be more appropriately placed in the methods section. In addition, much of the information stated can be extracted from the tables.
2. While it is interesting to see that MILL2 can adopt open conformations within the crystalline environment, whereas H2-Db cannot (and MIC proteins can exist in the open conformation without $\beta 2m$), the authors should show that H2-Db and MILL2 exhibit these differences outside of the crystallographic environment. Solution studies such as SAXS or other methods of determining size/shape characteristics could help elucidate if this open conformation for MILL2 exists in solution. These would be helpful and would relieve concern that the open conformation is an artifact of crystal-packing between the $\alpha 3$ and $\alpha 1$ - $\alpha 2$ domains between two neighboring MILL2 molecules (clearly visible in the PDB). The claims that $\beta 2m$ forms a tighter interaction with H2-Db as compared to MILL2 should be supported with experimental evidence.
3. The authors claim MILL2 does not bind peptide or present ligands however there is not definitive proof of this, so this point should be toned down. Heavy chains of peptide binding MHC molecules have been refolded successfully in previous studies without $\beta 2m$ or peptide, therefore the text should be modified to acknowledge these points and instead state that it is speculated that MILL2 binds no peptides or ligands in the groove.
4. It is interesting that MILL2 binds to the cellular surface of NIH-3T3 cells, presumably through HS and HSPGs. To validate this interaction, additional measurements should be performed, such as heparin affinity chromatography, SPR, or other biophysical techniques. Additionally, the various

mutants of MILL2 should also be tested this way. It is critical that other MHC Class I molecules be tested in these assays to verify that only MILL2 is binding the cellular surface (particularly H2-Db, but potentially other proteins as well). This would data would substantially improve the results.

5. The greatest area for improvement for this manuscript would be the inclusion of a functional assay to test MILL2/HS/HSPG interactions. The authors suggest that MILL2 enhances wound healing, therefore an appropriate wound healing assay testing the MILL2/HS/HSPG association would substantially enhance the impact of the structure and findings. Testing the various MILL2 mutants for reduced ability in a physiological assay would verify this point. In addition, while HS and HSPGs could be ligands for MILL2, perhaps it is also possible that these molecules are only functioning to localize MILL2, similar to that seen in many other signaling systems, such as TGF-beta, FGF, Wnt, etc. Perhaps MILL2 is utilizing HS and HSPGs to enhance binding to a target ligand/receptor for its functionality.

6. In figure 2 (d, e, and f), it would be much clearer to show these structures from the side and sliced such that the pockets of the α 1- α 2 can be more easily visualized and compared for their ligand pocket size. This would reveal differences between MILL2 and peptide binding molecules much better. Furthermore, perhaps adding MICA to this comparison would be beneficial, as it has no known ligand within this pocket and could enhance the claims that MILL2 also does not sequester ligands.

Reviewer #2:

Remarks to the Author:

Kajikawa et al. present the crystal structure of mouse MILL2, a non-classical MHC I molecule. Surprisingly, the authors found that the α 1- α 2 superdomain does not as tightly interact with the underlying α 3/ β 2M region and thus, the authors can both find an "open" and "closed" conformation in the crystal. In addition, the α 1- α 2 helices are located very close to each other, similar to MIC-A, suggesting that MILL2 does not present any peptide or other small molecule ligands. Surprisingly, the authors identify a basic patch in the α 3 domain as a binding site for the glycosaminoglycan heparan sulfate (HS) and suggest that MILL2 plays a role in wound healing through interaction with the proteoglycan.

The experiments are technically sound and include the appropriate controls. Data interpretation is appropriate and the figures are very clear.

The major concern is the presentation and discussion of the data. Firstly, the introduction is at times very vague (see comments below), also the authors do not discuss their findings sufficiently enough to raise sufficient interest in the presented work. What is the possible role of the "open" versus "close" state? Is it simply an representation of the domain flexibility of MILL2? Since the α 3 domain binds HS, what is the function of the α 1- α 2 domain? I think the manuscript is rather descriptive.

Minor comments:

1) crystal structure :

- a) The final PDB validation report needs to be submitted (not the preliminary report)
- b) Consider improving RSRZ outliers (81 in total)
- c) Too many close contacts

2) They authors refer to different functions of non-classical MHC I molecules twice in a row without ever mentioning any examples.

Line 64f.

"However, nonclassical MHC-I have diverse biological functions, which are not limited to cytosolic peptide antigen presentation to cytotoxic CD8+ T cells. Although some nonclassical MHC-I molecules bind peptides in the groove formed by the α 1- α 2 domains, they have roles which are not just restricted to the peptide presenting function of classical MHC-I".

The discussion of the different functions of nonclassical MHC I molecules and the ligands they bind is very vague. This should be presented more clearly and more precise. Name the ligand (lipids and microbial metabolites) and state the molecules (e.g CD1 and MR1).

Discussion:

There is some redundancy in the discussion related to the inability of MILL2 to bind ligands between the α 1- α 2 helices. eg.

line 254ff. "Secondly, the space between the two helices in the α 1- α 2 domains was too narrow to permit binding to peptides or small ligand molecules which typically associate with MHC-I molecules (Fig. 2d)."

And line 260ff. "Similar to the human MIC family, which also do not bind to small ligands, the two helices of the α 1- α 2 domains in MILL2 are too tightly spaced to permit binding to small molecules".

Reviewers' comments:

Reviewer #1 (Remarks to the Author):

We thank the reviewer for providing constructive comments.

Kajikawa and colleagues present data describing the structure of a newly described MHC Class I-like protein, MILL2. MILL2 is only found in mice, marsupials, and odd-toed ungulates and has no known homologue in humans, but is described as being most similar to the MIC MHC-like family of protein in humans. MILL2 has no known ligands, functions in a peptide and TAP independent manner, and has been suggested to play roles in wound healing and metabolism. However, the physiological roles of the MILL family of proteins is largely unknown. In this manuscript, the authors show that MILL2 forms a complex with $\beta 2m$ but can adopt two separate structures, a closed, classical MHC-like structure and an open conformation. This open conformation is similar to that seen for MICA and MICB. However, unlike these proteins, MILL2 is bound to $\beta 2m$.

Extending from the structure, the authors note the presence of a basic patch on the surface of MILL2 that localizes between both the $\alpha 1$ - $\alpha 2$ and $\alpha 3$ domains. The authors then show in the supplement that this basic patch is unique to MILL2, where other MHC Class I and Class I-like proteins typically maintain a more acidic nature here. From here, the authors then hypothesize and test whether or not MILL2 can bind heparan sulfate (HS) and HSPGs found on the cellular surface of NIH-3T3 cells using cell sorting analyses. These results are then extended, showing that heparinase as well as mutagenesis of this basic patch in MILL2 can abrogate the ability of MILL2 to bind to the cellular surface. As a result, the authors suggest that HSPGs may function as ligands for MILL2 and may play a role in its physiological function and wound healing.

Overall, the paper is significant in that a structure of a newly classified MHC Class I-like molecule is resolved and determines a new interaction of this protein with HS and HSPGs. While these findings are significant and worthy of publication, additional work

needs to be performed to push the impact of these findings for publication in Nature Communications. Suggestions for improvements (minor and major) are outlined below.

1. Starting on page 6 and through page 7 (lines 106 through 122), the details of the crystallographic methods is described in some details. This information would be more appropriately placed in the methods section. In addition, much of the information stated can be extracted from the tables.

Following the reviewer's comment, detailed explanation about crystallographic information has been deleted from the results section in the revised manuscript (Line 114-118).

2. While it is interesting to see that MILL2 can adopt open conformations within the crystalline environment, whereas H2-Db cannot (and MIC proteins can exist in the open conformation without β_2m), the authors should show that H2-Db and MILL2 exhibit these differences outside of the crystallographic environment. Solution studies such as SAXS or other methods of determining size/shape characteristics could help elucidate if this open conformation for MILL2 exists in solution. These would be helpful and would relieve concern that the open conformation is an artifact of crystal-packing between the α_3 and α_1 - α_2 domains between two neighboring MILL2 molecules (clearly visible in the PDB). The claims that β_2m forms a tighter interaction with H2-Db as compared to MILL2 should be supported with experimental evidence.

Following the reviewer's comment, we performed SAXS analysis for MILL2 revealing that the experimental scattering curve agreed well with the theoretically calculated scattering curve of the closed conformation (new Supplementary Fig. 4). This suggests that the majority of MILL2 in solution adopts the closed conformation, which is supposed to be more stable than the open one. On the other hand, interestingly, we performed heparin affinity chromatography (see the detail below) and found that only β_2m -free MILL2 binds to heparin, even though β_2m -conformed MILL2 was loaded (new Fig. 4). This suggested that MILL2 dissociates from β_2m to directly bind to heparin, likely via the open conformation. Therefore, although the closed conformation is advantageous for stable expression, the open conformation facilitates strong binding

to heparan sulfate by promoting release of β_2m . (In addition, even though many crystal structures of β_2m -conformed MHC class I molecules have been reported, these β_2m -conformed MHCIs only show the closed configuration.)

3. The authors claim MILL2 does not bind peptide or present ligands however there is not definitive proof of this, so this point should be toned down. Heavy chains of peptide binding MHC molecules have been refolded successfully in previous studies without β_2m or peptide, therefore the text should be modified to acknowledge these points and instead state that it is speculated that MILL2 binds no peptides or ligands in the groove.

Following with reviewer's comment, we modify the corresponding sentences in the revised manuscript as follows, (1) Line 98-99 “we speculate that MILL2 presumably plays no role in ligand presentation at the cell surface.” (2) Line 283-284 “We therefore speculate that, similar to the human MIC family, MILL2 does not bind peptides or small ligands in the groove formed by the α_1 - α_2 domains.”

4. It is interesting that MILL2 binds to the cellular surface of NIH-3T3 cells, presumably through HS and HSPGs. To validate this interaction, additional measurements should be performed, such as heparin affinity chromatography, SPR, or other biophysical techniques. Additionally, the various mutants of MILL2 should also be tested this way. It is critical that other MHC Class I molecules be tested in these assays to verify that only MILL2 is binding the cellular surface (particularly H2-Db, but potentially other proteins as well). This would data would substantially improve the results.

We performed heparin affinity chromatography using HiTrap Heparin HP column (GE healthcare) (new Fig. 4). As expected, wild type MILL2 protein bound to the column and eluted with 410 mM NaCl. On the other hand, the R194A/R200A/R251A mutant, whose tetramer does not bind to the cell surface of NIH 3T3 cells, hardly bound the column. Any small quantities of bound mutant MILL2 eluted at the lower concentration of NaCl (130 mM) compared to wild type MILL2. In addition, we also demonstrated that HLA-G, which is a human MHC-I molecule lacking the basic patch found in MILL2, could not bind to heparin column. These results strongly suggest that MILL2

binds to heparin directly by the basic patch on the $\alpha 3$ domain and binding to heparin/heparin sulfate is a unique function of MILL2.

5. The greatest area for improvement for this manuscript would be the inclusion of a functional assay to test MILL2/HS/HSPG interactions. The authors suggest that MILL2 enhances wound healing, therefore an appropriate wound healing assay testing the MILL2/HS/HSPG association would substantially enhance the impact of the structure and findings. Testing the various MILL2 mutants for reduced ability in a physiological assay would verify this point. In addition, while HS and HSPGs could be ligands for MILL2, perhaps it is also possible that these molecules are only functioning to localize MILL2, similar to that seen in many other signaling systems, such as TGF-beta, FGF, Wnt, etc. Perhaps MILL2 is utilizing HS and HSPGs to enhance binding to a target ligand/receptor for its functionality.

[REDACTED] following the suggestion of the reviewer, we added the sentence that “MILL2 would likely play a role in the recruitment of MILL2-expressing cells to fibroblasts by utilizing HS and HSPGs , in a way that is similar to many other signaling systems, such as TGF- β , FGF, Wnt, etc.” in the Discussion section (Lines 353-355). We also change the final sentence of the abstract as follows, “These findings suggest that MILL2 has a unique structural architecture and physiological role, binding to heparan sulfate proteoglycans on fibroblasts to possibly regulate cellular recruitment in biological events.”. We will continue to develop functional studies including in vivo study to define the physiological role of MILL2 in the future.

[REDACTED]

6. In figure 2 (d, e, and f), it would be much clearer to show these structures from the side and sliced such that the pockets of the $\alpha 1$ - $\alpha 2$ can be more easily visualized and compared for their ligand pocket size. This would reveal differences between MILL2

and peptide binding molecules much better. Furthermore, perhaps adding MICA to this comparison would be beneficial, as it has no known ligand within this pocket and could enhance the claims that MILL2 also does not sequester ligands.

We appreciate the reviewer's comment. In the revised manuscript, the figure is replaced with a cross-section of the structure of the $\alpha 1$ - $\alpha 2$ domains (Fig. 3d, e, f).

Reviewer #2 (Remarks to the Author):

We thank the reviewer for providing constructive comments.

Kajikawa et al. present the crystal structure of mouse MILL2, a non-classical MHC I molecule. Surprisingly, the authors found that the $\alpha 1$ - $\alpha 2$ superdomain does not as tightly interact with the underlying $\alpha 3/\beta 2M$ region and thus, the authors can both find an “open” and “closed” conformation in the crystal. In addition, the $\alpha 1$ - $\alpha 2$ helices are located very close to each other, similar to MIC-A, suggesting that MILL2 does not present any peptide or other small molecule ligands. Surprisingly, the authors identify a basic patch in the $\alpha 3$ domain as a binding site for the glycosaminoglycan heparan sulfate (HS) and suggest that MILL2 plays a role in wound healing through interaction with the proteoglycan.

The experiments are technically sound and include the appropriate controls. Data interpretation is appropriate and the figures are very clear.

The major concern is the presentation and discussion of the data. Firstly, the introduction is at times very vague (see comments below), also the authors do not discuss their findings sufficiently enough to raise sufficient interest in the presented work. What is the possible role of the “open” versus “close” state? Is it simply an representation of the domain flexibility of MILL2? Since the $\alpha 3$ domain binds HS, what is the function of the $\alpha 1$ - $\alpha 2$ domain? I think the manuscript is rather descriptive.

We performed SAXS analysis for MILL2. This analysis revealed that the closed conformation predominates over the open conformation in solution (also see the response for comments 4 and 5 of reviewer 1). On the other hand, heparin affinity chromatography also revealed that MILL2 binding to heparin promotes β_2m dissociation, indicating that MILL2 changes conformation from closed to open forms upon heparin binding. Hence, flexibility between closed and open conformations may facilitate HS binding to the basic patch in the $\alpha 3$ domain of MILL2. These points are discussed in lines 268-277 of the discussion.

Minor comments:

1) crystal structure :

a) The final PDB validation report needs to be submitted (not the preliminary report)

b) Consider improving RSRZ outliers (81 in total)

c) Too many close contacts

We now attach the final PDB validation report after improving the structure and re-submitting. RSRZ outliers and contacts have been improved, however, because of the poor electricity of the C-chain (MILL2 open form), a few violations are observed.

2) They authors refer to different functions of non-classical MHC I molecules twice in a row without ever mentioning any examples.

Line 64f.

“However, nonclassical MHC-I have diverse biological functions, which are not limited to cytosolic peptide antigen presentation to cytotoxic CD8+ T cells. Although some nonclassical MHC-I molecules bind peptides in the groove formed by the $\alpha 1$ - $\alpha 2$ domains, they have roles which are not just restricted to the peptide presenting function of classical MHC-I”.

We correct this redundancy in accordance with the reviewer's comment. Lines 62-64 read “Although some nonclassical MHC-I molecules display peptides in grooves formed by their $\alpha 1$ - $\alpha 2$ domains, these molecules also have additional roles which are not just restricted to the peptide presenting function of classical MHC-I to T cells⁸”

The discussion of the different functions of nonclassical MHC I molecules and the ligands they bind is very vague. This should be presented more clearly and more precise. Name the ligand (lipids and microbial metabolites) and state the molecules (e.g CD1 and MR1).

Following the reviewer's comment, in the revised manuscript, we rewrite this point more clearly. Line 64-70 "HLA-E binds peptide derived from the leader sequence of classical MHC-I for cell surface expression and interacts with CD94/NKG2 receptors on NK cells and T cells^{9,10}. In addition, other nonclassical MHC-I molecules bind low-molecular-weight non-peptide ligands in the grooves between the $\alpha 1$ - $\alpha 2$ domains, which, in many cases, determine their unique function. Microbial vitamin B metabolite bound to MR1 molecules activates mucosal-associated invariant T (MAIT) cells^{11,12}. The CD1 molecule family presents glycolipids to $\alpha\beta$ T cells or NKT cells¹³."

Discussion:

There is some redundancy in the discussion related to the inability of MILL2 to bind ligands between the $\alpha 1$ - $\alpha 2$ helices. eg.

line 254ff. "Secondly, the space between the two helices in the $\alpha 1$ - $\alpha 2$ domains was too narrow to permit binding to peptides or small ligand molecules which typically associate with MHC-I molecules (Fig. 2d)."

And line 260ff. "Similar to the human MIC family, which also do not bind to small ligands, the two helices of the $\alpha 1$ - $\alpha 2$ domains in MILL2 are too tightly spaced to permit binding to small molecules".

We correct this redundancy in accordance with the reviewer's comment. Line 283-284 "We therefore speculate that, similar to the human MIC family, MILL2 does not bind peptides or small ligands in the groove formed by the $\alpha 1$ - $\alpha 2$ domains." (see the response for above comment 3 of the reviewer 1)

Reviewers' Comments:

Reviewer #1:

Remarks to the Author:

Kajikawa and colleagues have resubmitted their manuscript on the structural and functional characterization of MILL-2, an MHC Class-I like protein found in mice, marsupials, and odd-toed ungulates with no known homologues in humans. The authors have addressed the major and minor concerns of this reviewer in detail and all substantial concerns have been addressed (as described in the reply to the reviewers). While more functional and physiological relevance would have been a nice supplement, it is certainly outside of the scope of this current manuscript. With these things in mind, this reviewer believes that the manuscript by Kajikawa and colleagues is acceptable in its current state for publication in Nature Communications.

Overall, this reviewer feels this manuscript is significant and important and adds new and necessary knowledge to our understanding of MHC-like proteins and the variability seen between these molecules.

Reviewer #2:

Remarks to the Author:

The manuscript has been improved and is now acceptable for publication